# Best Practices and Lessons Learned on Synthetic Data

**Ruibo Liu,**[*] **Jerry Wei,**[†] **Fangyu Liu**
Google DeepMind, Anthropic[†]

**Chenglei Si, Yanzhe Zhang**
Stanford University, Georgia Institute of Technology

**Jinmeng Rao, Steven Zheng, Daiyi Peng**
Google DeepMind

**Diyi Yang**
Stanford University

**Denny Zhou, Andrew M. Dai**[*]
Google DeepMind

## Abstract

The success of AI models relies on the availability of large, diverse, and high-quality datasets, which can be challenging to obtain due to data scarcity, privacy concerns, and high costs. Synthetic data has emerged as a promising solution by generating artificial data that mimics real-world patterns. This paper provides an overview of synthetic data research, discussing its applications, challenges, and future directions. We present empirical evidence from prior art to demonstrate its effectiveness and highlight the importance of ensuring its factuality, fidelity, and unbiasedness. We emphasize the need for responsible use of synthetic data to build more powerful, inclusive, and trustworthy language models.

## 1 Introduction

The rapid advancement of artificial intelligence (AI) technologies has led to their widespread adoption across numerous domains, from assistant agents (e.g., ACT-1, from Adept AI[1]) and software development (e.g., Devin, from Cognition Lab[2]) to healthcare (Singhal et al., 2022) and finance (Zheng et al., 2022). However, the success of AI models heavily relies on the availability of large, diverse, and high-quality datasets for training and evaluation. Acquiring such datasets can be a significant challenge due to data scarcity (Babbar & Schölkopf, 2019), privacy concerns (Abay et al., 2019), and the sheer cost of data collection and annotation (Gilardi et al., 2023a). Pessimists predict that we will run out of fresh text data in 2050 and image data in 2060 (Villalobos et al., 2022).

Synthetic data has emerged as a promising solution to address these challenges (Nikolenko, 2021). Synthetic data refers to artificially generated data that mimics the characteristics and patterns of real-world data, but is created through algorithms (Saxton et al., 2019), generative models (Borisov et al., 2022; Meng et al., 2022), or even simulations (Vezhnevets et al., 2023; Liu et al., 2023c), rather than being directly created by humans. By leveraging synthetic data, we can not only overcome the limitations of real-world data but also unlock

---

[*]Corresponding author(s): *ruiboliu@google.com, adai@google.com*
[†]Work done at Google DeepMind

[1]ACT-1: `https://www.adept.ai/blog/act-1`
[2]Devin: `https://www.cognition-labs.com/introducing-devin`

the potential to develop more robust, reliable, and fair AI models (Lucini, 2021; Lu et al., 2023).

One of the many benefits of synthetic data is that it can be generated at scale, providing an abundant supply of training and testing data for AI models. This is particularly valuable in domains where real-world data is scarce or difficult to obtain (e.g., weather data covering all conditions (Li et al., 2023a; Lam et al., 2023)). Second, synthetic data can be tailored to specific requirements, such as ensuring a balanced representation of different classes by introducing controlled variations (e.g., up-weighting low-resource languages in multilingual language learning (Przystupa & Abdul-Mageed, 2019)). This level of control over data characteristics can improve model performance and generalization. Third, synthetic data can help mitigate privacy concerns by creating anonymized or de-identified datasets that do not contain sensitive personal information (Howe et al., 2017; El Emam et al., 2020). This is crucial in domains such as healthcare, where patient privacy is of utmost importance (Dahmen & Cook, 2019; Wei et al., 2019).

Despite its promise, synthetic data also presents challenges that need to be addressed. One of them is ensuring the factuality and fidelity of synthetic data (Wood et al., 2021; Heusel et al., 2017), as models trained on false, hallucinated or biased synthetic data may fail to generalize to real-world scenarios (Van Breugel et al., 2023; Guarnera et al., 2020). Researchers must develop sophisticated generative models and evaluation metrics to create synthetic data that accurately reflects the complex patterns and relationships found in real-world data. Another challenge is the potential for synthetic data to amplify biases or introduce new biases if not carefully designed and validated (Barbierato et al., 2022; Gupta et al., 2021). We believe rigorous testing and fairness assessments are necessary to mitigate these risks.

In this paper, we track the current state of synthetic data research and discuss current best practices and lessons learned. The rest of the paper is organized as follows. Section 2 provides an overview of synthetic data generation techniques and their applications in model training, presenting case studies and empirical evidence. Section 3 discusses the usefulness of synthetic data in evaluation. Section 4 discusses the challenges and limitations of synthetic data, and in Section 5 we outline potential solutions and future research directions.

## 2 Synthetic Data in Training

Synthetic data, which is generated by mimicking authentic data collected from the real world, has proven to be an effective and relatively low-cost alternative of real data. This section explores several notable domains that leverages synthetic training data.

### 2.1 Reasoning

**Math.**  Recent advancements in mathematical reasoning for language models (LMs) have led to the development of various approaches to improve performance on math-related tasks. One approach is to train on math-targeted pre-training data, such as Minerva (Lewkowycz et al., 2022), Llemma (Azerbayev et al., 2023), and DeepSeekMath (Shao et al., 2024). Another mainstream method is to generate synthetic questions and answers to imitate the training or validation set of target benchmarks. For instance, WizardMath (Luo et al., 2023a) leverages a series of operations to increase the complexity of questions and answers using GPT-3.5, while MetaMath (Yu et al., 2023) bootstraps the questions in MATH and GSM8K by rewriting them in different ways, such as semantic rephrasing, self-verification, and backward reasoning. GAIR-Abel (Chern et al., 2023) found that the format of the augmented answers is crucial to final performance, with answers that begin with a paraphrasing of the question followed by a step-by-step solution showing better performance than those in vanilla format. Xwin-Math (Li et al., 2024) further scaled up synthetic SFT data to one million examples and found that the LLaMA-2 7B model (Touvron et al., 2023) can still benefit from data scaling. MMIQC (Liu & Yao, 2024) composed a bundle of datasets that infuse SFT style data (via question-answer rephrasing or directly taken from MetaMath) with a subset of high-quality mathematical pre-training data, such as OpenWebMath (Paster et al., 2023).

Scaling up the generation of synthetic math data is a straightforward process, but ensuring the correctness of the generated math remains a significant challenge for practitioners. AlphaGeometry (Trinh et al., 2024) is a recent attempt to address this issue by training a neural model using 100 million synthetic data points. The model proposes solutions and guides a symbolic deduction engine in verifying the correctness of each branch when solving complex geometry problems. By combining the power of synthetic data with a rigorous verification process, AlphaGeometry achieves a problem-solving ability comparable to that of a human Olympiad gold medalist, demonstrating the potential of this approach in tackling complex mathematical reasoning tasks.

**Code.** Different from Math, synthetic data for code reasoning can naturally combine the execution results with structured code, as one requirement of correct code is being executable. In coding-enhanced models, CodeRL (Le et al., 2022) presents an actor-critic approach to improve pretrained language models with feedback signals on synthetic code samples. Haluptzok et al. (2022) propose a self-improvement strategy where the models generate their own synthetic puzzle-solution pairs. These pairs are then verified and filtered by a real interpreter before being used to finetune language models. Shypula et al. (2023) further propose a framework that leverages a simulated environment and adaptation strategies like self-improvement synthetic data generation and CoT prompting for code optimization. Yang et al. (2024) developed InterCode, a framework designed to enhance interactive code generation within a reinforcement learning environment, where code serves as actions and execution feedback serves as observations. Reflexion (Shinn et al., 2024) employs external or internally simulated linguistic feedback signals to improve the code reasoning capabilities of language models. Regarding synthetic SFT data, Code Alpaca comprises a dataset of 20K code instructions automatically generated by applying SELF-INSTRUCT (Wang et al., 2022a) to ChatGPT across 21 seed tasks. WizardCoder (Luo et al., 2023b) introduces Code Evol-Instruct to guide ChatGPT with heuristic prompts to enhance the complexity and diversity of synthetic data. Meanwhile, Magicoder (Wei et al., 2023c) developed OSS-INSTRUCT, which generates 75K diverse synthetic instruction samples from open-source code snippets.

**Other reasoning tasks.** Synthetic data also leads to impressive performance in other reasoning tasks. For instance, Wei et al. (2023a) augmented existing natural language datasets by replacing natural language labels with arbitrary symbols, generating over 500k synthetic examples. Using these synthetic data for supervised finetuning significantly improved model performance on unseen in-context learning and algorithmic-reasoning tasks. STaR (Zelikman et al., 2022) generates synthetic chain-of-thought rationales and filters out those leading to wrong answers for finetuning language models to improve their reasoning. In the domain of physics reasoning, Mind's Eye (Liu et al., 2022) takes a novel approach by training a text-to-code model with synthetic "text-description → rendering code" data. This enables the model to convert textual questions into rendering code, which is then executed in a physical engine (i.e., DeepMind MuJoCo (Todorov et al., 2012)). The rendering results are injected into the context, allowing even small language models armed with Mind's Eye to achieve performance comparable to models 100 times larger.

## 2.2 Tool-using and Planning

**Learning tool-using through synthetic trajectories.** Synthetic data is also a powerful approach to enable LMs to learn tool-using abilities through simulated trajectories, as collecting real-world human tool-using data might be time-consuming, and the actual distribution of calls to tools might be skewed. LaMDA (Thoppilan et al., 2022), for instance, was trained not only on web documents but also on interaction data between crowdworkers and the model itself, with the synthetic data annotated with calls to appropriate tools. This training process allowed LaMDA to develop the ability to use a calculator for arithmetic, a search engine for real-time information seeking, and a machine translator for translation. Similarly, Toolformer (Schick et al., 2024) learns to decide which APIs to call and what arguments to pass by training on template-generated data, while Galactica (Taylor et al., 2022) infuse API-calling data into pre-training mixture. ToolAlpaca (Tang et al., 2023) is a novel framework designed to automatically generate a diverse tool-use corpus, by building a

multi-agent simulation environment and letting agents select and use tools iteratively. These examples demonstrate the potential of synthetic trajectories in enabling LMs to acquire tool-using abilities and enhance their reasoning capabilities across various domains.

**Learning to plan in synthetic environments.** An important feature of the agent in Autonomous Machine Intelligence (LeCun, 2022) is planning—an ability of decomposing complex tasks into subtasks and finishing the subtasks in a reward-optimal way (Kambhampati et al., 2024). Synthetic data can be a valuable tool here as it can serve as the feedback signal collected from a simulator (Park et al., 2023), and learning on it can make the agent aware of affordances (Ahn et al., 2022; Liang et al., 2022). For example, Inner Monologue (Huang et al., 2022) leverages natural language form feedback generated by the simulated environment to teach LLM-based robots planning. They find that such feedback significantly improves high-level instruction completion on both simulated and real-world domains. To compose a large number of realistic planning tasks (e.g., *"Rearrange objects on a table to match a given scene."*), VIMA (Jiang et al., 2022) creates a multi-modality simulated environment called VIMA-Bench, which supports extensible collections of objects and textures. In the Minecraft game, Voyager (Wang et al., 2023) deploys a number of GPT-4 based agents to interact with the synthetic environment and finds that the agents can unlock new skills faster and complete planning more efficiently with the help of synthetic feedback.

## 2.3 Multimodality

**Reverse rendering from vision to text.** Vision-language alignment data focuses on accurately grounding visual input to an LLM (usually via a vision encoder). Web-scraped image-caption pairs have been the most popular MM alignment data in the past few years since CLIP (Radford et al., 2021) and ALIGN (Jia et al., 2021). However, web-scraped image-text pairs are usually noisy and only have coarse-grained correspondence, insufficient for grounding details of images in language. In domains such as documents, screens, figures, and diagrams, such fine-grained alignment can most conveniently be obtained from data synthesis pipelines built with image rendering engines. Pix2Struct (Lee et al., 2023) uses web servers to render HTML code into website screenshots, and the training task is to derender a masked screenshot to the full HTML code. MatCha (Liu et al., 2023b) and DePlot (Liu et al., 2023a) render tabular data into charts with Python plotting libraries and pretrain a foundation model by giving the rendered image and producing the code and/or the tabular data. Si et al. (2024) and Laurençon et al. (2024) train on synthetically generated HTML and CSS files for the task of converting webpage screenshots into code implementation. The models finetuned on the synthetic data can generalize reasonably well on realistic data scraped from the Internet. Borkman et al. (2021) propose to use physics engines or game engines (e.g., Unity) as the synthetic data generator to help computer vision research.

**Multi-modality instruction following.** Downstream applications of multimodal LLMs require reasoning and instruction following capabilities. Such data are usually long-form question response pairs and are expensive for humans to create. LLaVA (Liu et al., 2024b) uses existing image captions to prompt GPT-4 (in text-only mode) for writing diverse and long-form prompt-answer pairs. During multimodal LLM training, images and prompts are used as input while the captions and bounding box information can be hidden. Besides image captions, other sources of image attribute information such as object bounding box (Zhao et al., 2023), OCR (Zhang et al., 2023c) and derendered charts (Masry et al., 2023; Carbune et al., 2024) can all fit into such as image attributes + text LLM rewriting synthetic data pipeline.

## 2.4 Multilingual

**Back-translation augmentation.** Many multilingual language models use back-translation as a data augmentation method, creating synthetic parallel training data from monolingual data sources (Sennrich et al., 2016; Zheng et al., 2020; Caswell et al., 2019; Marie et al., 2020; Bi et al., 2021; Liao et al., 2021; Pham et al., 2021; Xu et al., 2022). For example, Sennrich et al. (2016) back-translate monolingual target data into source language data, providing

additional parallel training samples for substantial translation task improvements. Researchers have also explored different sampling methods for back-translation (e.g., beam search, constrained sampling, unconstrained sampling) and their comparative effectiveness (Sennrich et al., 2016; Edunov et al., 2018; Graça et al., 2019; Bannard & Callison-Burch, 2005). Xu et al. (2022) emphasize the importance of the weight and quality of synthetic data for optimal NMT performance using back-translation. They propose a method to optimize the ratio between search methods and a gamma score to balance estimated importance weight and quality. However, some limitations exist with back-translation-based synthetic data generation. For example, the quality and diversity of synthetic data depends on the performance of the back-translation method. If the synthetic data is too noisy or not diverse, the performance gain would be limited (Epaliyana et al., 2021; Chauhan et al., 2022).

**Generating multilingual questions and answers at scale.** Recent studies explore the generation and utilization of synthetic multilingual question-answer (QA) pairs to improve language models' performance in multilingual and cross-lingual question answering (Asai et al., 2021; Kumar et al., 2019; Chi et al., 2020; Riabi et al., 2021; Li & Callison-Burch, 2023; Abulkhanov et al., 2023). One approach is to translate existing monolingual questions and/or answers into other languages (Asai et al., 2021). Another involves using Question Generation (QG) models to produce synthetic questions in a cross-lingual fashion based on answers and/or source texts (Kumar et al., 2019; Chi et al., 2020; Riabi et al., 2021). Recent efforts also focus on jointly generating questions and answers in multiple languages for greater flexibility (Shakeri et al., 2021; Li & Callison-Burch, 2023). For example, Shakeri et al. (2021) finetune a pretrained multilingual T5 model (Xue et al., 2020) on a mixture of a QA generation task and a multilingual masked language modeling task to produce synthetic QA pairs in multiple languages. These efforts generally show that language models trained on synthetic QA pairs demonstrate improved performance on multilingual QA and information retrieval benchmarks.

## 2.5   Alignment

**Instruction Following.** Synthetic data can serve as a promising approach for training instruction-following models, particularly in scenarios where real-world data is scarce, expensive, or challenging to obtain. Self-instruct (Wang et al., 2022a) and Stanford Alpaca (Taori et al., 2023) are both using LLMs to generate instruction following data which covers a wide range of scenarios. They first pick a small set of "seed instruction following samples" and then ask the LLMs to imitate the format to generate more demonstrations. One concern of this type of method is how to keep the generated data high quality, which involves the complexity of queries (Liu et al., 2023d), the diversity of semantics (Ding et al., 2023), and the scale of the synthetic dataset (Yuan et al., 2023). To this end, Xu et al. (2023) propose Evol-Instruct which adds complexity to simple instructions via prompting. Mukherjee et al. (2023) leverage LLMs to revise the instructions and responses iteratively to include high-quality explanation traces in the FLAN dataset (Wei et al., 2022), and they find the trained model has improved performance in many NLP tasks. UltraChat (Ding et al., 2023) is large-scale and multi-round synthetic dialogue dataset, which is generated by two separate ChatGPT Turbo API models—one serves as the user role while the other serves as the assistant. They instruct the user model with carefully designed prompts to mimic real human user behaviors.

Many language models are supervised finetuned to learn how to follow instructions, but in learning this behavior, they may inadvertently also learn to be *sycophantic* (Perez et al., 2023), tailoring their responses to follow a user's viewpoint, even if that viewpoint is not objectively correct (Wei et al., 2023b). Sharma et al. (2024) find evidence that the preference models (i.e., the reward model used for RLHF training) and even humans prefer sycophantic responses sometimes. On this front, Wei et al. (2023b) generates synthetic data to encourage models to be robust to user opinions and adds these data in a finetuning step to reduce sycophantic behavior on held-out prompts.

**Mitigating hallucination.** Many widely-used language models utilize supervised finetuning (SFT) to learn to align their interactions with users (Wang et al., 2022b; Zhang et al.,

2023b). In particular, there exist many methods of generating synthetic SFT data that can improve capabilities such as reasoning and alignment (Wei et al., 2023a;b). It has been shown, however, that these synthetic data can induce hallucinations into language models by containing nontrivial amounts of hallucinated answers or by forcing models to learn to answer questions that they do not know the answer to (Zhang et al., 2023d). These cases demonstrate that synthetic data, if not applied correctly, can actually increase hallucinations in language models.

On the other hand, recent work has also shown promising results in mitigating hallucinations using synthetic data. For example, GPT-4 (OpenAI, 2023) was trained using a reward model that leveraged synthetic hallucination data in order to perform reinforcement learning (Zhang et al., 2023d). This method resulted in a significant improvement in performance on the TruthfulQA (Lin et al., 2022) dataset (Zhang et al., 2023d). Similarly, Jones et al. (2023) designed a synthetic task where hallucinations can be readily evaluated, utilizing this task to optimize LLM outputs by learning a continuous postfix via prefix-tuning. Tian et al. (2023) uses automated fact-checking and confidence scores to rank factuality scores of model response pairs, which are then used to finetune language models with DPO (Rafailov et al., 2023) to improve their factuality. Continued research in using synthetic data to mitigate hallucinations is still limited, however, by the lack of synthetic tasks for which hallucinations can be scalably evaluated.

**Aligning with shared human preference and values.** Directly finetuning on value-aligned or human-preferred data is a straightforward method for aligning language models, but this method often requires substantial human annotation, which can be prohibitively expensive at scale. Additionally, such annotation frequently exhibits varying styles and inconsistent quality, particularly in the case of poorly annotated samples at the lower end of the quality spectrum (Meta, 2023; Gilardi et al., 2023a). To address these practical challenges, an advanced technique known as "reinforcement learning from human feedback (RLHF)" has been proposed (Leike et al., 2018; Christiano et al., 2017; Ouyang et al., 2022). This approach involves training a reward model with human data to act as a proxy of human judgment, which guides the optimization of the LM generation policy.

Recent studies have proposed a mixture of synthetic data and real human data to train more robust reward models (Gao et al., 2023). Constitutional AI (Bai et al., 2022) proposes to use a small set of principles to steer the AI generated critiques and feedback, and use such synthetic data to replace the real human data in the typical RLHF pipeline. The model trained with this RLAIF (i.e., reinforcement learning from AI feedback) method shows similar strong performance as RLHF baselines. In general, synthetic data offers a powerful solution for human values and preferences alignment by allowing researchers to generate large-scale, diverse, and controlled training datasets in a low-cost way (Cui et al., 2023; Ganguli et al., 2022). By simulating a wide range of scenarios involving ethical dilemmas (Perez et al., 2022), social interactions (Liu et al., 2023c), and cultural norms (Ziems et al., 2023), synthetic data enables comprehensive and systematic testing of AI models' alignment with human values (Askell et al., 2021). This approach helps identify and mitigate issues related to bias (Liu et al., 2021; Ntoutsi et al., 2020), fairness (Zhao et al., 2018; Landers & Behrend, 2023), and unintended consequences before AI systems are deployed in real-world settings (Ye et al., 2024).

However, it is important to acknowledge that low-fidelity synthetic human preference data might be limited in accurately reflecting nuanced human judgment (Argyle et al., 2023). Consequently, the resulting models may be less robust under "jail-breaking attacks" (Huang et al., 2023a; Deshpande et al., 2023), and may reveal strategically deceptive behavior even through safety training (Pan et al., 2022; Steinhardt, 2022; Everitt et al., 2021). To mitigate these risks, researchers must continuously refine and improve the quality and diversity of synthetic data, incorporating more complex and comprehensive scenarios that better capture the intricacies of human values and preferences. Additionally, combining synthetic data with real-world data, and creating synthetic data in an interactive environment which can be synced with the real world, are promising remedies. As the need for effective AI governance and regulation grows, synthetic data will play an increasingly vital role in enabling

scalable oversight mechanisms that promote trust, accountability, and the development of AI technologies that are aligned with human values and societal expectations.

## 3 Synthetic Data in Evaluation

Synthetic data is widely used in evaluations of different perspectives:

**Factuality.** AI systems may generate information or responses that are not grounded in factual knowledge or data, leading to the creation of misleading or false content, formally known as *hallucination* (Ji et al., 2023). Factuality evaluation aims to ensure the consistency of the knowledge in the AI system's output with the knowledge provided by its training data and knowledge base (Ji et al., 2023; Zhang et al., 2023d). Early statistical-based hallucination evaluation methods relied on n-grams to directly calculate the overlap of vocabulary between the input and output content (Dhingra et al., 2019; Wang et al., 2020). However, these methods have limitations, as they only consider lexical overlap and do not account for semantics or sentence meaning (Ji et al., 2023), making them unsuitable for evaluating more complex forms of hallucination. Subsequent assurance methods shifted from statistical approaches to model-based methods, which are more robust compared to token-difference-based methods (Honovich et al., 2021). While these model-based evaluation methods are more advanced than their predecessors, they still have limitations. For example, the models can only output the degree of hallucination and may struggle to pinpoint specific errors (Falke et al., 2019). Feng et al. (2023a) propose to combine LLMs generation with random walks on knowledge graphs to generate synthetic evaluation data for factuality, which is aware of entities and relations on the graphs. Wei et al. (2024) created a synthetic dataset called LongFact for long-form factuality evaluation and used Google Search as the grounding source and LLM for the automated judgement, to achieve human-level accuracy but with significally lower cost (Min et al., 2023).

**Safety.** Red teaming is a powerful technique for evaluating the safety and robustness of AI models (Ganguli et al., 2022; Casper et al., 2023b). By generating diverse and realistic scenarios designed to elicit unaligned or harmful outputs (Casper et al., 2023a), red teaming can expose vulnerabilities and weaknesses in AI systems (Perez et al., 2022). For example, Perez et al. (2023) use LMs to generate datasets for evaluating the behavior of other LMs. They end up producing 154 high-quality datasets which are verified by humans, and discover new cases of inverse scaling where LMs get worse with size. Hubinger et al. (2024) leverage synthetic data to trigger backdoor attacks to LMs at scale; they find LMs can exhibit deceptive behavior and create a false impression of safety under such attacks, and standard "safety training" could not remove such deception easily. These methods demonstrate the feasibility of using AI assistance to scale up human oversight (Bowman et al., 2022) over complex problems and unseen domains.

**Assisting human evaluation.** Recent studies have shown that in many cases, synthetic judgements from large-scale LMs (LLMs) can serve as qualified, fast, and low-cost alternatives to actual human evaluation (Gilardi et al., 2023b). Using GPT-4 as the judge, Alpaca Eval (Li et al., 2023b) and MT Bench (Zheng et al., 2023) are two popular benchmarks that measure the comprehensive abilities of LM-based ChatBot. In coding tasks, synthetic environment is a common choice to aid human evaluation, as humans can make the assessment more efficiently via actual executions and analysis on running logs. Gu et al. (2024) propose CRUXEval, a code execution reasoning benchmark consisting of 800 Python functions generated by CodeLLaMA-34B. Similarly, Liu et al. (2024a) introduce CodeMind, a framework to gauge the code reasoning abilities of LLMs on Independent Execution Reasoning (IER), Dependent Execution Reasoning (DER), and Specification Reasoning (SR). All these evaluations based on synthetic data show strong correlation with real human judgements.

# 4 Challenges and Limitations of Synthetic Data

While synthetic data offers numerous benefits and applications, it is crucial to acknowledge and address the potential challenges and limitations associated with its use. This section delves into three significant concerns surrounding synthetic data:

**Misuse of synthetic data might proliferate misinformation.** The potential misuse of synthetic data is a significant concern that must be addressed to ensure the responsible development of AI systems. Current AI models become increasingly capable of generating human-like data ranging from text (Gemini-Team et al., 2024; 2023), images (Saharia et al., 2022; Ramesh et al., 2022), songs [3], to even videos (e.g., OpenAI SORA [4]). This can be particularly dangerous when synthetic data is used to impersonate real people, manipulate public opinion, or influence political processes. Moreover, the dissemination of synthetic data-driven misinformation can erode trust in legitimate information sources, making it increasingly difficult for people to distinguish between truth and falsehood (Byman et al., 2023; Rid, 2020). To mitigate these risks, it is crucial for researchers, developers, and policymakers to establish clear guidelines and best practices for the ethical generation and use of synthetic data, including robust mechanisms for detecting and countering synthetic misinformation (Groh et al., 2022). By proactively addressing these challenges, we can harness the benefits of synthetic data while minimizing its potential for harm.

**Synthetic data might cause ambiguity in AI alignment.** The increasing use of synthetic data in aligning AI models (e.g., Constitutional AI (Bai et al., 2022)) can introduce significant ambiguity and uncertainty. The goal of AI alignment is to ensure that AI systems behave in ways that are aligned with human values and intentions. However, synthetic data, which is artificially generated rather than collected from real-world sources, may not accurately represent the nuances and complexities of human values and preferences (Zhou et al., 2024). This discrepancy can lead to AI models learning from data that is biased (Feng et al., 2023b; Liu et al., 2021), ungrounded (Liu et al., 2022; Patel & Pavlick, 2022), or misrepresentative of real-world scenarios (Weidinger et al., 2021; Ji et al., 2023). As a result, AI systems trained on synthetic data may exhibit behaviors that are misaligned with human expectations, potentially leading to unintended consequences or even harmful actions (Zou et al., 2023; Anderljung et al., 2023). Moreover, the ambiguity introduced by synthetic data can make it challenging to interpret and understand the decision-making processes of AI models (Lightman et al., 2023), further complicating the task of ensuring alignment. To mitigate these risks, it is crucial for researchers to carefully consider the limitations and potential drawbacks of using synthetic data in alignment research and to develop robust methods for validating and testing AI models trained on such data.

**Training with synthetic data makes evaluation decontamination harder.** The use of synthetic data in model training poses significant challenges to fair evaluation. Evaluation benchmarks are often created by referring to public text sources, such as coursework websites or forums. Consequently, it is arguable that all publicly available benchmark test cases might occasionally be included in the pre-training data of LLMs (Hoffmann et al., 2022; Gao et al., 2021). The use of synthetic data exacerbates this issue rather than mitigating it. Although the community has proposed several techniques to detect such evaluation contamination, such as *min-k% prob* (Shi et al., 2023), which checks the probabilities of $k$ long-tail tokens, these token-level decontamination methods are inadequate when the model is trained with synthetic data. Synthetic data might include rephrased versions of the benchmark data (Oren et al., 2023; Mattern et al., 2023), rendering token-level decontamination ineffective. In addition to developing more advanced evaluation contamination detection techniques, we recommend that model developers invest in creating and maintaining in-house and protected evaluation benchmarks. These proprietary benchmarks should be carefully safeguarded to prevent leakage and ensure the integrity of the evaluation process.

---

[3]Make songs with Suno AI: `https://app.suno.ai/`
[4]OpenAI Sora: `https://openai.com/research/video-generation-models-as-world-simulators`

# 5 Directions for Future Work

As the field of synthetic data continues to evolve, there are several promising directions for future research and development. This section outlines three key areas that warrant further exploration:

**Synthetic data scaling.** The impressive performance of many over-trained small language models (e.g., Mistral series models (Jiang et al., 2023), and Gemma series models (Gemma-Team et al., 2024), *inter alia*) demonstrates the necessity of training with large amount of tokens (even passing the compute-optimal chinchilla law (Rae et al., 2021)). However, whether we have similar conclusions on the training with synthetic data is still an open question, as the quality of synthetic data may not be as consistent as real-world data (Yu et al., 2024). Future research should investigate the scaling laws for synthetic data and determine the optimal balance between the quantity and quality of synthetic samples. This exploration could help us understand the most effective strategies for leveraging synthetic data in training large-scale language models, potentially leading to more efficient and cost-effective approaches (Muennighoff et al., 2024).

**Further improving quality and diversity of synthetic data.** While existing methods for generating synthetic data have shown promise, there is still room for improvement in terms of creating high-quality, attributed synthetic samples that closely mimic real-world data. Future research should focus on developing new advanced techniques (or based on existing ones such as Generative Adversarial Networks (GANs) (Goodfellow et al., 2020) or Diffusion Models (Ho et al., 2020), *inter alia*) that can control and manipulate specific attributes of the generated data, enabling the creation of diverse and customizable synthetic datasets. Additionally, researchers should explore methods that can incorporate domain-specific knowledge to ensure the generated data adheres to the underlying constraints and patterns present in the target domain (e.g., via Retrieval Augmented Generation (RAG) (Lewis et al., 2020; Borgeaud et al., 2022)) while maintaining the data quality. By advancing the state-of-the-art in attributed synthetic data generation, we can unlock new opportunities for privacy-preserving analysis (Assefa et al., 2020), and model training across various fields, from healthcare (e.g., synthetic medical images (Frid-Adar et al., 2018; Wei et al., 2019)) and finance (e.g., simulated trading trajectories (Zheng et al., 2022)) to social sciences (Argyle et al., 2023; Park et al., 2023) and beyond.

**Towards high-fidelity and more efficient scalable oversight.** As AI models become increasingly complex and autonomous, it becomes challenging to monitor and assess their behavior using traditional oversight methods that rely on human supervision or real-world data (Amodei et al., 2016). Future research should explore the use of synthetic data for high-fidelity scalable oversight of these advanced systems. Existing methods typically simulate a certain scenario in social iterations, such as debate (Leike et al., 2018), reflection (Zhang et al., 2023a), or revisions (Liu et al., 2023c) to obtain synthetic data, while new approaches could cover more comprehensive scenarios and more modalities (Sun et al., 2023), as recent studies have found many issues of simulation that only covers a narrowed down (Cheng et al., 2023) or over-simplified (Zhou et al., 2024) scenes. Looking forward, another growing direction could be how to achieve scalable oversight more efficiently—given that we have the full control over the synthetic data generation, we can probably provide more targeted oversights with less synthetic data. As the need for effective AI governance and regulation grows, synthetic data will play an increasingly vital role in enabling more trustworthy scalable oversight mechanisms that promote robust, accountable, and safe deployment of AI technologies for the benefit of society (Askell et al., 2021; Bowman et al., 2022).

**The emergent self-improvement capability.** We typically choose the most capable model to generate synthetic data, as its generation is of higher quality. However, an intriguing question arises: can a model generate synthetic data that is better than the data it was trained on, thus enabling it to improve itself? This concept of self-improvement through synthetic data generation is an exciting avenue for future research. If a model can generate higher-quality data than its original training set, it could potentially bootstrap its own performance by iter-

atively learning from the enhanced synthetic data (Chen et al., 2024). This self-improvement capability could lead to the emergence of more advanced AI systems that can autonomously refine their skills and knowledge over time (Burns et al., 2023; Huang et al., 2023b). Although recent work shows encouraging progress in this direction (Chen et al., 2024; Yuan et al., 2024), the upper bound of self-improvement and the underlying reasons for its effectiveness remain open questions. Future research should investigate the theoretical underpinnings and practical feasibility of self-improvement through synthetic data generation in more diverse scenarios, examining the necessary conditions, potential limitations, and associated risks. By unlocking the potential of emergent self-improvement capabilities, we could enable more adaptable, efficient, and autonomous learning processes (LeCun, 2022).

## 6 Conclusion

Synthetic data has emerged as a promising solution to address the challenges of data scarcity, privacy concerns, and high costs in AI development. By generating realistic and diverse datasets, synthetic data enables the training and evaluation of AI models at scale across various domains. As we approach human-level or even superhuman-level intelligence, obtaining synthetic data becomes even more crucial, given that models need better-than-average-human quality data to progress. However, ensuring the factuality, fidelity, and lack of bias in synthetic data remains a critical challenge.

Future research directions on synthetic data could focus on improving the fidelity and controllability of generative models and developing standardized evaluation and contamination protocols and tools. We could also explore the integration of synthetic data with other techniques and its application in other domains. Despite the challenges, the potential benefits of synthetic data in advancing AI research are significant. By leveraging synthetic data responsibly and effectively, we can build more powerful, inclusive, and trustworthy AI systems that benefit society as a whole.

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
