# OpenReview forum: "Best Practices and Lessons Learned on Synthetic Data"
_colmweb.org/COLM/2024/Conference — COLM_

### Official Review · Reviewer_XqGy · 2024-04-28

**Rating:** 7
**Confidence:** 4
**Ethics Flag:** 1

**Summary:**

This survey paper presents a comprehensive summary of the uses of synthetic data, issues with its quality, and possible research directions that are relevant to explore in the realm of developing and utilizing synthetic data.
The discussion of applications where synthetic data is used and can benefit is very thorough and gives an almost complete view of the ways researchers generate and use synthetic data nowadays in training models, the section about the use of synthetic data for evaluation is less detailed since it doesn't refer to the fact that data that is synthesized to train models is also being used to evaluate the models by using some sort of partition to train, test and dev datasets.
The last two sections are a good starting point for researchers to examine when they consider developing synthetic data and/or working with it, though I thought that some of the points there are obvious and may need to be explained from a different perspective (for example, how improving the quality of synthetic datasets different than improving the quality of non-synthetic ones). The part about the self-improving models through synthetic dataset creation is very interesting though.

**Reasons To Accept:**

* A helpful survey for researchers interested in creating synthetic data or developing methods for generating synthetic data.
* Some of the future research directions mentioned in the paper are thought-provoking.

**Reasons To Reject:**

If this was not a summary paper, I wouldn't write this comment as a reason to reject the paper, but since it is then some of the points mentioned in the paper are obvious and the authors could have focused more on the less-known issues with synthetic data, and raise more questions in the future research direction section.

---

> ### Author Rebuttal · Authors · 2024-05-31
>
> Thanks for your insightful reviews!
>
> We agree with the reviewer that it would be better to include additional discussions on the less-known issues of synthetic data, and raise more open questions for further discussion. Specifically, we plan to add more discussion on the privacy protection of synthetic data, and potential techniques that can mitigate model collapse if the synthetic data is not carefully produced. We hope these additions can further improve our paper!

---

> > ### Comment · Reviewer_XqGy · 2024-06-05
> > **Acknowledgment of Rebuttal**
> >
> > Thank you for your reply. I encourage you to make these changes and add the discussion to the paper.

---

### Official Review · Reviewer_gT5a · 2024-05-07

**Rating:** 5
**Confidence:** 3
**Ethics Flag:** 1

**Summary:**

This paper surveys recents results in the topic of synthetic text data used to train and evaluate machine learning algorithms such as large langage models (LLM) for natural language and code. The results are grouped in topics such as use-case and type of tasks on which synthetic data is being used to enhance performance. A number of limitations are discussed and several directions for future work are proposed.

The work provides a useful survey of how synthetic text data is being used. However, I believe that the paper would benefit from better scoping, articulating its methodology, and applying a critical lens to the field to nuance some of the « hyped »  statements.

**Questions To Authors:**

Questions:

- How does your work relate to other surveys in this space, e.g., https://arxiv.org/abs/2205.03257?

Main comments:

- Please consider adding formal definitions and corresponding terms for the various types of synthetic data considered in this work, such as template-generated, simulated, algorithm-generated synthetic data, etc.

- The statement that synthetic data can create « anonymized » datasets « that do not contain sensitive personal information » is incorrect. An important missing reference is Stadler et al. [A] which demonstrates that synthetic data is not immune to inference attacks (such as membership inference and attribute inference), one criteria for anonymisation as per the Working Party 2014’s Opinion on Anonymization techniques. The notion that synthetic data is inherently privacy-preserving has been seriously challenged (see also Houssiau et al, 2022 [B] and Meeus et al., 2023 [C]). Unless formal privacy protections are put in place (differential privacy guarantees), synthetic data will leak private information, especially about vulnerable records. How to achieve synthetic data that is truly privacy-preserving while remaining useful is an open problem.

- How much of the literature in this space relies on template/algorithm-generated synthetic data, with prior knowledge infused by humans and how much of it uses LLM-generated synthetic data ? When is it better to use one over the other?

- Please consider adding a table with the datasets which are publicly available. This could help this paper become a useful starting point for identifying already existing synthetic text datasets.

- The paper surveys mainly results from the last couple of years, however synthetic data similar to the focus of this work has already been used in the past, see e.g., Kocijan et al.[D]’s dataset for solving the Winograd Schema Challenge. Please consider including citations from older work and incorporating their contribution in the review.

- The question asked in the «emergent self-improvement capability » paragraph seems to have already been (negatively) answered in Shumailov et al. [E] Please consider including a discussion of this paper and updating the conclusions in light of their finding.

[A] Stadler, T., Oprisanu, B., & Troncoso, C. (2022). Synthetic data–anonymisation groundhog day. In 31st USENIX Security Symposium (USENIX Security 22) (pp. 1451-1468).

[B] Houssiau, F., Jordon, J., Cohen, S. N., Daniel, O., Elliott, A., Geddes, J., ... & Szpruch, L. (2022, October). TAPAS: a Toolbox for Adversarial Privacy Auditing of Synthetic Data. In NeurIPS 2022 Workshop on Synthetic Data for Empowering ML Research.

[C] Meeus, M., Guepin, F., Creţu, A. M., & de Montjoye, Y. A. (2023, September). Achilles’ heels: vulnerable record identification in synthetic data publishing. In European Symposium on Research in Computer Security (pp. 380-399). Cham: Springer Nature Switzerland.

[D] Kocijan, V., Cretu, A. M., Camburu, O. M., Yordanov, Y., & Lukasiewicz, T. (2019, July). A Surprisingly Robust Trick for the Winograd Schema Challenge. In Proceedings of the 57th Annual Meeting of the Association for Computational Linguistics (pp. 4837-4842).

[E] Shumailov, I., Shumaylov, Z., Zhao, Y., Gal, Y., Papernot, N., & Anderson, R. (2023). The curse of recursion: Training on generated data makes models forget. arXiv preprint arXiv:2305.17493.

**Reasons To Accept:**

- The paper provides a useful survey of use cases/tasks on which synthetic text data has been successfully used to train/evaluate ML algorithms. Thus, the paper can serve as a reference for readers interested in understanding this space and in re-using existing synthetic text datasets similar use cases.

**Reasons To Reject:**

- The scope of the paper is not clear from the Title, Abstract and Introduction. The paper presents itself as a survey of synthetic data in general, but its scope is more narrow: synthetic text data mainly generated using hand-engineered templates and algorithms (but not generative algorithms). This is not a bad thing : a review of synthetic text data can still be very useful. However, clarifying the scope is essential to reaching the intended audience, because what most people understand by « synthetic data » is artificial data generated by an AI model, e.g., a GAN or a Bayesian network, which was trained on real data.

- The paper is very extensive on the applications of synthetic text data, but is missing a taxonomy/categorization of the synthetic data generation models. The introduction does not delineate the boundary between human-aided techniques (such as algorithms/simulations) and fully data-driven synthetic data (« rather than being directly created by human. » ). A table of available synthetic datasets grouped by method and/or use case is missing.

- The methodology used to identify the tasks to which synthetic text data is applied is not clear. It is not clear if the review is exhaustive of the field, and how « good » use cases are delineated from « bad » use cases.

---

> ### Author Rebuttal · Authors · 2024-05-31
>
> Thanks for your thoughtful reviews and constructive feedback! We are glad our paper is useful in providing a comprehensive reference for future research on synthetic data for LLMs. Below we address your questions:
>
> **Q1: Adding formal definition of synthetic data**
>
> Yes we will have a better definition and categorization of different types of synthetic data (mainly categorized by the generation approach) in our final version.
>
> **Q2: Privacy concerns on synthetic data**
>
> Thanks for the great question and many useful references! We agree with the reviewer that whether synthetic data can is still an open question --- our original claim was mainly based on some prior art that claimed synthetic data can mitigate the concerns about personal privacy (e.g., Harnessing large-language models to generate private synthetic text). In our final version we will leave the room for further discussion on this and will include the work you mentioned as references for different opinions. Thanks again for the thoughtful question!
>
> **Q3: Current situation of template-based synthetic data generation**
>
> Although now generating synthetic data with LLMs seems to become the dominant method, we still see in some applications template-based generation can still play a role. For example, Google DeepMind used sympy + python to create millions of synthetic arithmetic QAs to aid the LM to gain better mathematical abilities [1]. In general there seems to be no clear boundary; it more depends on the performance of the method on specific applications.
>
> **Q4: Adding tables of public available synthetic datasets**
>
> Great suggestion! We will add one in our final version.
>
> **Q5: Adding more related work in the early years**
>
> Thanks for the suggestion! It is indeed important to cite some related work in early years so that our readers can better understand the past and present of synthetic data research. We will include more in our next version!
>
> **Q6: Adding more discussion**
>
> Thanks for the suggestion! We agree with the reviewer that how to make sure training on synthetic data would not cause “model collapse” is still a challenge for synthetic data research. We will add more discussion on this in our final version.
>
> Finally, we really appreciate the reviewer’s thoughtful comments and constructive suggestions! Hope our rebuttal can solve your concerns and questions. Thanks!
>
> [1] [Analysing Mathematical Reasoning Abilities of Neural Models](https://github.com/google-deepmind/mathematics_dataset)

---

> > ### Comment · Reviewer_gT5a · 2024-06-04
> >
> > Thank you for your reply. The reply addresses the concerns raised in the "Main comments" part of my review and I agree with the changes proposed by the authors. Unfortunately, my main concerns (cf Reasons to Reject and Question) have not been addressed in the authors' response.

---

### Official Review · Reviewer_5Caz · 2024-05-09

**Rating:** 8
**Confidence:** 3
**Ethics Flag:** 1

**Summary:**

The paper presents a survey of existing methods on synthetic data research, and its various applications. The paper discusses synthetic data generation techniques across various tasks like reasoning, multilinguality, multi-modality, etc. Next, the paper discusses the use of synthetic data in evaluations. Finally, challenges and future directions for research in this area are discussed. The paper is well-written and easy to follow.

**Questions To Authors:**

- Typographical errors:\
    - Page 2. Para 2: ‘’synthetic’’ -> ‘’synthetic data’’
    - Page 6: Missing citation under ‘’Aligning with shared human preference and values’’
- Some additional literature:
    1. Section 2.4:
        - Aya Dataset: An Open-Access Collection for Multilingual Instruction Tuning (Singh et al., 2024)
    2. Section 2.5:
        - SelfCheckGPT: Zero-Resource Black-Box Hallucination Detection for Generative Large Language Models (Manakul et al., EMNLP 2023)
    3. Section 4:
        - The Generative AI Paradox: "What It Can Create, It May Not Understand" (West et al., 2024)

**Reasons To Accept:**

1. This work is a timely contribution as the NLP community is rapidly moving towards using synthetically generated data for scale.

2. The work comprehensively touches on various application areas and discusses a detailed list of literature for each of these areas. Each section is a great starting point for a future researcher looking to leverage automatically generated data.

**Reasons To Reject:**

1. While most of the literature reviewed in this paper understandably pertains to the the post-LLM era, synthetic data generation strategies have had a rich history even before the era of transformers. For instance, annotation projection [1,2], systematic perturbation[3,4], bi-/multi-linugal pivoting [5], to name a few. A section discussing these methods would make the work more complete.
2. The discussion on the consistency of LLM outputs (or the lack thereof) deserves to be mentioned in detail. Several works have shown inconsistency as a problem [6,7,8]

[1] Inducing Multilingual Text Analysis Tools via Robust Projection across Aligned Corpora (Yarowsky et al., HLT 2001) \
[2] Multilingual Projection for Parsing Truly Low-Resource Languages (Agić et al., TACL 2016)\
[3] The Galactic Dependencies Treebanks: Getting More Data by Synthesizing New Languages (Wang & Eisner, TACL 2016)\
[4] Synthetic Data Made to Order: The Case of Parsing (Wang & Eisner, EMNLP 2018)\
[5] Paraphrasing with Bilingual Parallel Corpora (Bannard & Callison-Burch, ACL 2005)\
[6] CONDAQA: A Contrastive Reading Comprehension Dataset for Reasoning about Negation (Ravichander et al., EMNLP 2022)\
[7] Consistency Analysis of ChatGPT (Jang & Lukasiewicz, EMNLP 2023)\
[8] Language models are not naysayers: an analysis of language models on negation benchmarks (Truong et al., *SEM 2023)

---

> ### Author Rebuttal · Authors · 2024-05-31
>
> Thanks for reading our work carefully and the high rating! We are glad you think our work is a timely contribution to the community! Below we address questions:
>
> **Q1: Typos**
>
> Yes we will of course fix these typos in our final version. Thanks again for reading our paper carefully!
>
> **Q2: Additional references**
>
> Thanks for the great suggestions on additional references! Since we are not allowed to update our paper during rebuttal, we will make sure we have them in our final version!
>
> Finally, thanks for the careful reading and great suggestions! Hope our work can help your future research!

---

> > ### Comment · Reviewer_5Caz · 2024-06-05
> >
> > Thank you for the response.

---

### Official Review · Reviewer_MjuL · 2024-05-23

**Rating:** 6
**Confidence:** 5
**Ethics Flag:** 1

**Summary:**

The paper provides best practices of synthetic data research from the lens of various tasks/case-studies/domains. It buckets the research  directions in the domains both from training and evaluation standpoint. It also describes the challenges of synthetic data generation research in the each of the domains, however falls short on the sufficient "empirical evidence" mentioned in the abstract.
The primary categories during training are:
* reasoning
* tool-using and planning
* multimodal
* Multilingual
* Alignment.

Whereas, the categories for same in evaluation are:
* hallucination
* safety
* assisting human evaluation.

**Questions To Authors:**

* (Possible) missing citation: ? in "Aligning with shared human preference ...." paragraph

* Reference to, and discussion on "Wang et al, Data Management For Large Language Models: A Survey" can help improve the paper

* It is slightly unclear why "mitigating" hallucination has been clubbed under alignment subsection.

* I *really* enjoyed the "directions of future work". Kudos!

**Reasons To Accept:**

* Provides necessary and important case-studies for usage of synthetic data that helps different LLM benchmarks.

* For every case-studies, the paper references the important synthetic data papers, which can serve as an interesting survey of some sorts.

**Reasons To Reject:**

* While the paper does provide various case-studies, it doesn't *"delve"* much into the empirical evidences on various techniques.

* There are various techniques highlighted for each case-study in the paper, it would be good to have some remarks on "best practices" on each of them at least on some empirical level.

* The crucial "take-home" message is missing for each domain.

---

> ### Author Rebuttal · Authors · 2024-05-31
>
> Thanks for your thoughtful reviews and great suggestions! We are glad to see you find our paper a great survey on important case-studies of using synthetic data. Below we address your questions:
>
> **Q1: Missing citation**
>
> Yes we will add the citation [1] to the proper position. Thanks for reading our paper carefully!
>
> **Q2: Referring to a prior art**
>
> Yes it is indeed an important work we can refer to. We will add related text and cite this work in our final version.
>
> **Q3: Why “mitigating hallucination” is under “Alignment” section**
>
> We are following some prior arts on the definitions: For example, in [2], the author defined the “mitigating hallucination” problem as one of the “Knowledge Alignment” problems.
>
> **Q4: Really enjoying the “direction of future work” section**
>
> Thanks! We are so glad you love reading this section! Hope it helps your future research!
>
> Finally, we thank the reviewer again for the insightful comments!
>
> [1] [Llama 2: Open Foundation and Fine-Tuned Chat Models](https://arxiv.org/abs/2307.09288)
>
> [2] [​The Knowledge Alignment Problem: Bridging Human and External Knowledge for Large Language Modelse](https://arxiv.org/abs/2305.13669)

---

> > ### Comment · Reviewer_MjuL · 2024-06-06
> > **Response to the rebuttal**
> >
> > Thank you for the bullet-wise responses.
> > A3 does make sense. Thank you!

---

### Decision · Program_Chairs · 2024-07-10

**Decision:**

Accept

**Comment:**

Overall, reviewers found the paper useful: they appreciate having a post-LLM era review paper, and felt this would be a good resource for the field. However, there were some points that didn't fully land with them, so they gave useful suggestions to help the paper reach its full potential. In particular, the paper's scope and contextualization should be clarified for the next version, as recommended by reviewers gT5a and 5Caz. In particular, attention should be paid to addressing changes since the last review paper mentioend by 5Caz, and including related work mentioned by all reviewers. It could also be nice to include a take-home message as suggested by reviewer MjuL, at the end of each section.

Everyone participated in rebuttal, and useful info was exchanged.